# Morphological and Molecular Characterization of the Benthic Dinoflagellate *Amphidinium* from Coastal Waters of Mexico

Lorena María Durán-Riveroll [1,2,*], Oscar E. Juárez [3], Yuri B. Okolodkov [4], Ana Luisa Mejía-Camacho [5], Fabiola Ramírez-Corona [6], Dania Casanova-Gracia [7], María del Carmen Osorio-Ramírez [8], Victor A. Cervantes-Urieta [9] and Allan D. Cembella [2,8,*]

1   CONAHCYT, Departamento de Biotecnología Marina, Centro de Investigación Científica y de Educación Superior de Ensenada, Ensenada 22860, Mexico
2   Alfred-Wegener-Institut, Helmholtz-Zentrum für Polar-und Meeresforschung, 27570 Bremerhaven, Germany
3   Programa de Acuicultura, Centro de Investigaciones Biológicas del Noroeste, S.C., La Paz 23096, Mexico
4   Instituto de Ciencias Marinas y Pesquerías, Universidad Veracruzana, Boca del Río 94294, Mexico
5   División de Ciencias Naturales e Ingeniería, Universidad Autónoma Metropolitana-Cuajimalpa, Mexico City 05348, Mexico
6   Taller de Sistemática y Biogeografía, Departamento de Biología Evolutiva, Facultad de Ciencias, Universidad Nacional Autónoma de México, Mexico City 04510, Mexico
7   Facultad de Ciencias Biológicas, Cd Universitaria, Benemérita Universidad Autónoma de Puebla, Puebla 72592, Mexico
8   Departamento de Biotecnología Marina, Centro de Investigación Científica y de Educación Superior de Ensenada, Ensenada 22860, Mexico
9   Programa de Doctorado en Recursos Naturales y Ecología, Facultad de Ecología Marina, Universidad Autónoma de Guerrero, Acapulco 39390, Mexico
*   Correspondence: lmduran@cicese.mx or lduran@conacyt.mx or lorena.duran@awi.de (L.M.D.-R.); allan.cembella@awi.de or cembella@cicese.mx (A.D.C.)

**Abstract:** The genus *Amphidinium* Clap. & J. Lachm. comprises a high diversity of planktonic and benthic (epiphytic and sand-dwelling) dinoflagellates from marine and freshwater ecosystems. High morphological plasticity and vaguely defined genus characteristics (e.g., a small epicone size) have complicated the clear delineation of species boundaries. Although six *Amphidinium* morphospecies have been reported from Mexican coastal waters, species identifications are uncertain and not generally supported by molecular phylogenetic data. In this study, seven isolates of *Amphidinium* from diverse benthic coastal locations on the NE Pacific, Gulf of California, and southern Gulf of Mexico were subjected to critical morphological analysis using photonic and scanning electron microscopy. The phylogenetic reconstruction was based on nuclear-encoded, partial large-subunit (LSU) rDNA and internal transcribed spacer I and II (ITS1 and ITS2) sequences. The revised phylogenetic analysis was consistent with the traditional subdivision of the genus *Amphidinium* into two sister groups: Herdmanii and Operculatum clades. This study provided the first confirmed records of *A. theodorei* and *A. massartii* from coastal waters of Mexico. The molecular phylogenetic evidence indicated that the morphologically described *A.* cf. *carterae* from Baja California was in fact more closely allied with *A. eilatiensis* sequences. A few *Amphidinium* species are known to form toxigenic (i.e., fish-killing) harmful algal blooms worldwide, and therefore knowledge on species diversity and biogeography is critical in developing effective strategies for evaluating the potential emerging threat in Mexican coastal waters.

**Keywords:** amphidiniales; benthic dinoflagellate; epiphyte; dinoflagellate phylogeny; large-subunit rDNA; MP; maximum parsimony; ML; maximum likelihood

## 1. Introduction

The genus *Amphidinium* Clap. & J. Lachm. emend. Flø Jørgensen, Murray & Daugbjerg, 2004 is a diverse group of athecate dinoflagellates found worldwide in aquatic environments. High morphological, trophic, and biological plasticity are noteworthy features of the

genus [1]. *Amphidinium* includes marine and freshwater species classified as heterotrophic or autotrophic, although all species are likely capable of at least limited mixotrophy. Both benthic and planktonic *Amphidinium* species are often found as members of their respective microeukaryote communities in coastal zones. The species may be exclusively epibenthic, sand-dwelling, endosymbiotic, or planktonic, whereas others are tychoplanktonic as occasional residents of the water column [2–4].

Historically, athecate forms with a small to minute epicone as compared with the size of the hypocone [5] or with an epicone of one-third or less of the total cell length [6] have been ascribed to *Amphidinium*. The genus *Amphidinium* was redefined based on multiple morphological features as follows: "Athecate benthic or endosymbiotic dinoflagellates with a minute irregular triangular-or crescent-shaped epicones. Epicone overlaying anterior ventral part of the hypocone. Epicone deflection to the left. Cells dorso-ventrally flattened, with or without chloroplasts" [2].

The type species *A. operculatum* was also redescribed based on photonic (light microscopy (LM)) and scanning electron microscopy (SEM) and combined with partial large-subunit (LSU) rDNA sequence data [4], which allowed the establishment of defined species boundaries. The cell shape is among the most important morphological features defined by the amphiesma of *A. operculatum*. The right margin of the hypocone is convex, whereas the left one is almost straight; the epicone overlays the anterior central part of the hypocone. The shape of the epicone is irregular triangular in the ventral view with the left anterior left tip deflected to the left, a slightly descending cingulum, and a narrow ventral ridge running between the two points of flagellar insertion.

Up to 2014, 29 benthic *Amphidinium* species had been described [3], including 6 *sensu lato* or whose generic affinity had not yet been investigated or confirmed. Within Algae-Base [7], there are now more than 90 taxonomically accepted *Amphidinium* species. As the taxonomic affinities are resolved (particularly with revisions based on molecular sequencing approaches), the list of benthic *Amphidinium* species will doubtless increase. The benthic *Amphidinium* species comprise slightly less than one-third of the total number known for this genus. In any case, scientific and public health interest in the global phylogeography and cell abundance has focused primarily on benthic *Amphidinium* species within the last decade. Putatively toxigenic *Amphidinium* and species that produce allelopathic and/or other biologically active polyketides are more heavily represented among the benthic than pelagic members of the genus [8]. Bottom-dwelling *Amphidinium* species may be associated with benthic harmful algal blooms (bHABs) [9]; the human health risk linked to bHABs is especially acute in tropical and subtropical regions dependent upon seafood resources.

Despite occasional *Amphidinium* blooms reported from Mexico [10–12], there are few records for the coastal subregions. In the absence of a systematic national HAB monitoring program, there is scarce information on *Amphidinium* blooms and risk assessment along the coasts of Mexico. Members of the genus are commonly found (often at high cell densities) in both pelagic and benthic habitats. In the southwestern Gulf of Mexico, *A.* cf. *carterae* reached an abundance of $41 \times 10^3$ cells g$^{-1}$ substrate wet weight upon the chlorophyte *Ulva fasciata* Delile [10]. In Bahía de La Paz, Gulf of California, a pelagic bloom of *A.* cf. *carterae* reached a density of $1 \times 10^6$ cells L$^{-1}$ [12], but no impacts on aquaculture, ecosystem function, or human health were reported.

*Amphidinium* blooms remain, however, a potential emerging problem posing an undefined risk to human health and seafood security on the Pacific coast of Mexico, the Gulf of California, and the southern Gulf of Mexico. A fish kill during an *Amphidinium* bloom was reported from a coastal lagoon near Sydney, Australia [13], but similar fish mortalities have not been circumstantially linked to *Amphidinium* in Mexico. Moreover, to date there are no confirmed cases of human poisoning in Mexico from consuming seafood contaminated by any known toxigenic *Amphidinium* species.

Several *Amphidinium* species, particularly certain strains of *A. carterae, A. gibbosum* (Maranda & Shimizu) Flø Jørgensen & Murray, *A. massartii* Biecheler, and *A. operculatum*, are considered "toxigenic" because they synthesize an array of bioactive polyketides, partic-

ularly macrolides and diverse short- and long-chain linear polyketides known informally as amphidinolides and amphidinols, respectively. Members of these polyketide families often share a core unit but differ in length and structure of the lateral chains, including amphidinols (AM), luteophanols, lingshuiols, symbiopolyols, karatungiols, carteraols, and related long-chain polyketides [14]. At this stage, it is premature to consider these polyketides as toxins because the specific toxicity of most of them against defined molecular targets remains undefined. Nevertheless, many of these *Amphidinium* polyketides have exhibited fungal and hemolytic bioactivity [15–17] and in some cases cytotoxic effects against certain cancer cell lines [18,19].

Despite the frequent occurrence of polyketides in cultured isolates of *Amphidinium* and their fish-killing potential in the laboratory, there is no direct confirmation that blooms of this genus have caused ecological damage or human health consequences attributable to their toxigenicity. Circumstantial indications that bHABs of *Amphidinium* may be linked to the multiple effects and etiology of the complex syndrome ciguatera fish poisoning (CFP) have not been confirmed. *Amphidinium* occurs frequently in high cell abundance in dinoflagellate assemblages with *Gambierdiscus* Adachi & Fukuyo and *Fukuyoa* Gómez, Qiu, Lopez & Lin, both known to produce potent ciguatoxins (CTX) and/or maitotoxins (MTX) [20], the primary causes of CFP. *Amphidinium* species are not known to produce polyether derivatives of either CTX or MTX, but the potent bioactive amphidinols could contribute synergistically to the CFP syndrome via food chain accumulation within fish.

The morphological plasticity within and among species and the absence of stable surface features due to the lack of thecal plates has led to frequent revisions and reassignments at the species level. Furthermore, taxonomic evaluation of *Amphidinium* specimens based exclusively on morphology may lead to misidentifications because of the similarities among species and the cellular plasticity at different stages of the *Amphidinium* life cycle [2,4,21]. The original species descriptions are frequently unreliable since many have not been micrographed and lack iconotypes and/or are based on fixed specimens and not on living cells. Fixation modifies the cell shape, leading to misidentification [22] because *Amphidinium* cells are extremely delicate and easily destroyed or deformed.

Many taxonomic and phylogenetic relationships within the genus are being resolved via the application of molecular phylogenetic approaches [1,2,21] by sequencing of the respective rRNA genes, including the LSU and internal transcribed spacer I and II (ITS1 and ITS2) regions. According to the molecular phylogenetic data on *Amphidinium*, the species have been subdivided into two sister clades (the Herdmanii clade and the Operculatum clade) as first proposed by Flø Jørgensen, Murray and Daugbjerg [2] and maintained in the subsequent literature [1].

To date, six species of *Amphidinium* have been reported from Mexico with SEM images provided for *A. theodorei* from Bahía de la Paz, Baja California Sur (BCS) (reviewed in [23]); only *A. carterae* and *A. operculatum* were previously illustrated [24]. Most of the current taxonomic record of *Amphidinium* species from the coastal waters of Mexico is based solely on morphological characterization. Molecular sequence data have been published only for two species (*A. operculatum* and *A. carterae*) [23]. The research herein presents the morphological identification of *Amphidinium* species from four regions of the coast of Mexico via LM and SEM. These identifications were compared and confirmed by using a molecular phylogenetic analysis of the variable D1-D6 region of the LSU and ITS1-ITS2 regions of the rRNA gene of cultured isolates of *Amphidinium* species. This study comprised the first records of *A. theodorei* and *A. massartii* from Mexico. The molecular phylogenetic evidence from these gene sequences indicated that the provisionally described *A.* cf. *carterae* from Baja California was in fact more closely allied with *A. eilatiensis* J.J. Lee, R. Olea, M. Cevasco, X. Pochon, M. Correia, M. Shpigel & J. Pawlowski. The knowledge contributed herein on the diversity and distribution of *Amphidinium* species is essential for designing an effective monitoring program for risk assessment and early warning systems for bHABs in Mexican coastal waters.

## 2. Materials and Methods

### 2.1. Sampling, Cell Isolation, and Culture of Amphidinium

Cells of *Amphidinium* species were isolated from Bahía de La Paz (BCS) (24°09′30.01″ N 110°19′12.10″ W), the Veracruz Reef System (VRS) (Veracruz, 19°11′54.10″ N, 96°4′0.70″ W), Laguna de Términos (Campeche, 18°38′18.72″ N, 91°46′14.73″ W), and San Quintín (Baja California, 30°27′14″ N, 116°00′15″ W) (Figure 1).

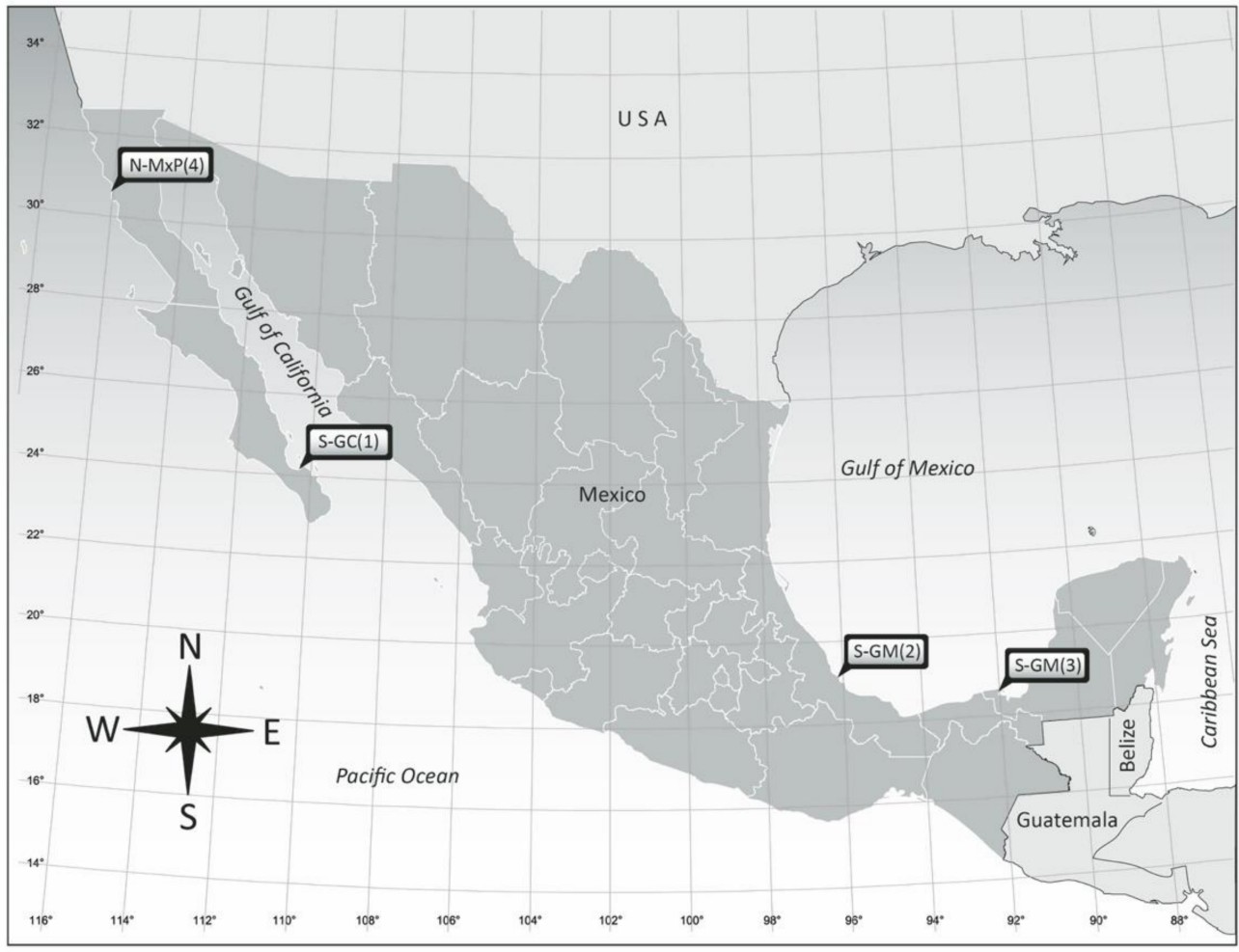

**Figure 1.** Localities of origin of the *Amphidinium* isolates studied from the coasts of Mexico. Geographical site location numbers are referenced in Table 1.

Samples were collected at intervals from January to June in 2018 and January and December in 2019. These *Amphidinium* specimens were collected from floating macroalgae *Sargassum* sp., *Padina* sp. (Phaeophyceae), and other macroalgae attached to buoys as well as seagrasses *Thalassia testudinum* Banks ex König and *Zostera marina* L. Single cells of epibenthic *Amphidinium* were isolated with a micropipette into sterile 96-well microplates containing 300 μL of 50%-strength GSe growth medium [25] modified without soil extract and prepared from autoclaved (121 °C, 15 min) seawater filtered through sand, activated carbon, and 1 μm cartridge filters. The growth medium (supplemented with $GeO_2$ (final concentration: 2.5 mg $L^{-1}$) [26] to inhibit diatom growth) was prepared from heat-sterilized seawater stock at a salinity of 36. Clonal isolates were cultured at 25 ± 1 °C on a 12:12 h light:dark photoperiod and under illumination of 50 μmol photons $m^{-2}$ $s^{-1}$.

**Table 1.** Provisional species assignment according to morphological characteristics of cultured clonal isolates of *Amphidinium* from coastal locations in Mexico indicating locality of origin and substrate type. BC = Baja California; BCS = Baja California Sur; S-GC = southern Gulf of California; S-GM = southern Gulf of Mexico; N-MxP = northern Mexican Pacific. Numbers in parentheses refer to location numbers in Figure 1.

| Isolate | Locality of Origin | Geographical Region | Substrate Type | Provisional Morphospecies |
|---|---|---|---|---|
| AcSAV105 | Veracruz Reef System, Veracruz | S-GM (2) | Anchored buoy | *A. carterae* Hulburt |
| AeSQ172 | San Quintín, BC | N-MxP (4) | *Zostera marina* | *A.* cf. *carterae* |
| AeSQ177 | San Quintín, BC | N-MxP (4) | *Z. marina* | *A.* cf. *carterae* |
| AeSQ181 | San Quintín, BC | N-MxP (4) | *Z. marina* | *A.* cf. *carterae* |
| AmLT112 | Laguna de Términos, Campeche | S-GM (3) | *Thalassia testudinum* | *A. massartii* Biecheler |
| AA60 | Veracruz Reef System, Veracruz | S-GM (2) | *Padina* sp. | *A. operculatum* Clap. & J. Lachm. |
| AtLPZ38 | Balandra, La Paz, BCS | S-GC (1) | Floating brown alga | *A. theodorei* C.R. Tomas & Karafas |

After a growth period of between 21 and 28 days, the contents of the microplate wells with the highest cell densities were transferred to 60 × 15 mm Petri dishes with full-strength (100%) GSe medium and assigned an isolate number (Table 1). Cultures for molecular characterization and morphological analysis were initiated from 15 mL inoculum in 250 mL Erlenmeyer borosilicate flasks with 125 mL of modified GSe medium. Cultures were maintained under illumination at 50 μmol photons m$^{-2}$ s$^{-1}$ on a 12:12 h light:dark photoperiod and at 23 ± 1 °C. Cell growth was monitored until maximal optical density via visual inspection. After two weeks of growth, 15 mL from each culture was harvested under sterile conditions for detailed LM and SEM analyses.

### 2.2. Cell Morphology and Statistical Analysis

For the LM, 50 μL of live cell culture was diluted 1:1 with filtered seawater. Cells were observed on an inverted (Axio Observer.A1, Carl Zeiss, Oberkochen, Germany) or vertical LM (Motic BA310E, Hong Kong SAR, China) equipped with phase-contrast optics at 200–1000× magnification. After visual inspection of cells, images were recorded with the following digital cameras: Axiocam 506 color camera (6 MP) (Carl Zeiss Microscopy GmbH, Göttingen, Germany) and Moticam S6 (6 MP) (Motic, Hong Kong SAR, China).

The SEM was performed following a specimen preparation method for delicate dinoflagellates [27]. Briefly, 15 mL of cultured *Amphidinium* cells were centrifuged at 3000× *g* (4 °C) for 5 min. Pelleted cells were fixed for 90 min with 750 μL osmium tetroxide (OsO$_4$) solution (2% final concentration) or 2% glutaraldehyde. Fixed samples were washed three times in 1.5 mL of 5 °C ultrapure water and centrifuged at 3000× *g* (4 °C) for 5 min. The cells then underwent a graded ethanol (EtOH) dehydration series (10, 20, 30, 40, 50, 60, 70, 80, 90, and 99%; 1.5 mL each). At every step, the cells were gently resuspended for 1 min and centrifuged under the previous conditions, and the supernatant was discarded. After 99% EtOH removal, 200 μL of hexamethyldisilazane:ethanol (HMDS:EtOH) (1:1 *v/v*) was added; cells were resuspended and centrifuged again. To complete the dehydration, 200 μL of pure HMDS was added, and the cells were gently resuspended for 1 min. Samples were placed on SEM stubs and left for air drying and then gold sputter-coated for 5 min. Cells were observed with a JSM 6360-LV scanning electron microscope (JEOL, Tokyo, Japan) equipped with a backscattered electron detector under 8 kV voltage acceleration and at a 15 mm working distance.

Statistical analyses and associated figures were developed with the R software and programming language (version 4.1.1) and in the RStudio integrated development environment (version 2022.02.0 + 443) [28]. Kruskal–Wallis non-parametric tests were applied to analyze the morphometric variability of all isolates with the *kruskal.test function*. When there were differences among variances, a Mann–Whitney–Wilcox post hoc test was applied

to determine the effect size for each comparison adjusted by the Bonferroni correction in the function *pairwise.wilcox.test* from the stats v.4.1.1 package. A value of $p < 0.05$ was chosen to indicate statistical significance.

### 2.3. Total Genomic DNA Extraction and Amplification

Cell cultures (30–50 mL each) were harvested via centrifugation (10 min at $3000 \times g$) (Solbat J12, Puebla, Mexico) at room temperature (21 °C) to obtain the cell biomass. The total genomic DNA of *Amphidinium* was extracted using the modified CTAB method [2,4,13] or with a Qiagen Dneasy PowerSoil kit (QIAGEN, Redwood City, CA, USA) following the manufacturer's recommendations. Briefly, for the CTAB method, the pellets and 1 mL of supernatant were transferred to a 1.5 mL microtube. To facilitate cell lysis, cells were frozen at −20 °C for 48 h and then thawed during extraction [29]. DNA was extracted following a protocol previously described [30] and stored in TBE buffer at −20 °C. The DNA concentration was measured in a NanoDrop spectrophotometer (Thermo Scientific, MA, USA) and purified with the DNA Clean & Concentrator kit (Zymo Research, CA, USA). Analysis of the DNA quantity was performed with a spectrophotometer (Epoch, Biotek, VT, USA) at absorbance wavelengths of A260 and A280 nm. For DNA extraction method using the Qiagen Dneasy PowerSoil kit, the cell biomass was placed in PowerBeat tubes (Qiagen, Redwood City, CA, USA), and 60 µL of C1 solution was added. The mix was incubated for 10 min at 70 °C. Then, samples were agitated in a FastPrep-24 tissue homogenizer (MP Biomedicals, Irvine, CA, USA) for 30 s at a velocity of 6 m s$^{-1}$ (three times) with a 3 min ice-bath pause between each step. The biomass was centrifuged for 10 min at $10,000 \times g$ at 21 °C. The DNA extraction was verified using agarose electrophoresis and quantified via microvolume spectrophotometry in a Nanodrop 2000 (Thermo Scientific, San Diego, CA, USA).

Amplification of the D1 and D6 variable domains of the LSU rRNA gene of the *Amphidinium* isolates (comprising approximately 1450 base pairs (bp)) was carried out using the primers D1R (5′-ACCCGCTGAATTTAATTTAAGCATA-3′) [31] and 28–1483R (5′-GCTACTACCACACCAAGATCTGC-3′) [32]. The polymerase chain reaction (PCR) was performed in a thermal cycler (Flexigene, Techne, Staffordshire, England, UK) with a Phire Plant Direct PCR Master Mix kit (ThermoScientific, MA, USA). The PCR conditions were as follows: an initial denaturation cycle at 98 °C for 5 min followed by a 20 s denaturation cycle at 98 °C and 40 cycles of alignment at 48 °C for 30 s, polymerization at 72 °C for 20 s, and a final extension of one minute at 72 °C. The PCR products were visually verified under UV light on 1% agarose gel stained with ethidium bromide (EtBr) after electrophoresis (90 V per 5 min).

The ITS1-ITS2 region amplification was performed with the primers ITSF2 (TACGTC-CCTGCCCTTTGTAC) and ITSR2 (TCCCTGTTCATTCGCCATTAC) [33] with a GoTaq colorless Master Mix kit (Promega, Fitchburg, Wisconsin, USA) according to the manufacturer's instructions. The amplification conditions for the PCR thermocycler (BIO-RAD Universal Hood III, BIO-RAD, Irvine, CA, USA) were: 95 °C for 2 min, 95 °C for 30 s, 35 cycles at 60 °C for 45 s, and finally 72 °C for 2.5 and 7 min.

### 2.4. Sequencing and Phylogenetic Analysis

PCR products were sequenced with a Genetic Analyzer ABI Prism 3100 (Applied Biosystems, CA, USA) by seqXcel, Inc (San Diego, CA, USA). The quality of the sequences was evaluated in 4Peaks v1.8 (Nucleics, Sydney, New South Wales, Australia), and only sequences of good quality (average quality $\geq 20$ on the PHRED scale) were considered for analysis. All sequences were checked in the NCBI database using BLASTn analysis to determine the species assignments. Phylogenetic analyses were focused on two different genomic regions. The first included the internal transcribed spacer 1, the 5.8 S ribosomal RNA, and the internal transcribed spacer 2 (ITS1–5.8S rRNA-ITS2); the second region consisted of the 28S large-subunit (LSU) ribosomal RNA gene operon. Available sequences corresponding to these regions from the *Amphidinium* species were retrieved from the Gen-

Bank nucleotide database (Table 2). The corresponding sequences from the distantly related dinoflagellate *Heterocapsa* sp. were used as an outgroup in the phylogenetic analyses [1,34]. Phylogenetic trees were constructed for each region separately, and then alternative trees were built for both concatenated regions.

**Table 2.** Taxa, strain ID numbers, GenBank accession numbers, number of base pairs (bp) in each sequence, and author references for all *Amphidinium* species sequences used in this study.

| | | ITS2 | | | | LSU | | |
|---|---|---|---|---|---|---|---|---|
| **Taxa** | **Strain** | **GenBank Acc.** | **bp** | **Authors \*** | **Strain** | **GenBank Acc.** | **bp** | **Authors \*** |
| *A. carterae* | DL1100 | FJ907458 | 636 | Zhao et al. | CS-740 | AY460578 | 1359 | Murray et al. |
| *A. eilatiensis* | CCMP2100 | AJ417900 | 576 | Lee et al. | CCMP2100 | AJ417900 | 886 | Lee et al. |
| *A. fijiensis* | Amfi0508-1 | KY697937 | 1053 | Karafas et al. | WZD19 | MZ351945 | 1304 | Luo et al. |
| *A. gibbosum* | D2A1 | MK418355 | 533 | Zhang et al. | CCMP 120 | AY455672 | 1325 | Jörgensen et al. |
| *A. magnum* | ZS606 | OM177176 | 506 | Xie, H. | Amma0206-1 | KY070341 | 1163 | Karafas et al. |
| *A. massartii* | NEPCC 802 | FJ823531 | 522 | Stern et al. | CCMP 1821 | AY455670 | 1338 | Jörgensen et al. |
| *A. operculatum* | SKLMP_S091 | MK605120 | 1499 | Yiu et al. | SM06 | AY455674 | 1376 | Jörgensen et al. |
| *A. paucianulatum* | Ampa0606-2 | KY697960 | 1056 | Karafas et al. | Ampa0606-1 | KY070345 | 1163 | Karafas et al. |
| *A. pseudomassartii* | AKLV01 | KY697945 | 1010 | Karafas et al. | AKLSPO1 | AY460588 | 1357 | Murray et al. |
| *A. steinii* | TIO181 | MZ359142 | 602 | Luo et al. | SM12 | AY460593 | 1426 | Murray et al. |
| *A. theodorei* | Amth0702-1 | KY697942 | 695 | Karafas et al. | Amth1303-1 | KY697986 | 1317 | Karafas et al. |
| *A. thermaeum* | Amth00600FA1 | KY697956 | 803 | Karafas et al. | Amth0304-1 | KY070365 | 1159 | Karafas et al. |
| *A. tomasii* | Amto1412-1 | KY974334 | 1127 | Karafas et al. | Amto1412-2 | KY697984 | 1270 | Karafas et al. |
| *Heterocapsa* sp. | NA | JN020158 | 640 | Zadabbas et al. | CCMP424 | AY371082 | 860 | De Salas, M. |
| *A. carterae* | AcSAV105 | OQ448886 | 559 | Present study | AcSAV105 | MT325892 | 949 | Present study |
| *A. eilatiensis* | AeSQ172 | OQ448892 | 596 | Present study | AeSQ172 | OQ435732 | 860 | Present study |
| *A. eilatiensis* | AeSQ177 | OQ448891 | 593 | Present study | AeSQ177 | OQ435731 | 857 | Present study |
| *A. eilatiensis* | AeSQ181 | OQ448890 | 590 | Present study | AeSQ181 | OQ435730 | 857 | Present study |
| *A. massartii* | AmLT112 | OQ448887 | 715 | Present study | AmLT112 | MT325893 | 926 | Present study |
| *A. operculatum* | AA60 | OQ448888 | 719 | Present study | AA60 | MT325891 | 874 | Present study |
| *A. theodorei* | AtLPZ38 | OQ448889 | 732 | Present study | AtLPZ38 | MT325890 | 813 | Present study |

\* Sequence author according to GenBank database. NA: information not available.

Independent alignments for each gene were performed with MUSCLE [35] and ClustalW [36] algorithms, verified by eye and trimmed in MEGA7, then used to perform all the phylogenetic analyses [37]. For both genes, two phylogenetic approaches—maximum likelihood (ML) and maximum parsimony (MP)—were compared (Supplementary Materials). Since the ML method is based on a nucleotide substitution model, MEGA7 was employed to find the substitution model that best fit each gene. In the ML method, initial trees for the heuristic search were obtained automatically by applying Neighbor-Joining and BioNJ algorithms to the matrix of pairwise distances estimated using the Maximum Composite Likelihood (MCL) and selecting the tree topology with a superior log likelihood value. A discrete Gamma distribution was used to model evolutionary rate differences among sites [37]. In comparison, the MP trees were obtained from the Subtree-Pruning-Regrafting (SPR) algorithm, in which the initial trees were generated by the random addition of sequences (10 replicates). Tree branch lengths were calculated using the average pathway method [38]. Both the ML and MP consensus trees were obtained after 2000 bootstrap replicates [39]. The optimal trees were selected after comparison for the best bootstrap branch support and congruence according to previous phylogenetic analyses of *Amphidinium* species [1,40] (Supplementary Materials). Since the MP method did not assume any nucleotide substitution model, distances were not represented in the MP tree, and only the topology is shown.

## 3. Results

### 3.1. Morphological Characterization

The morphological analyses of *Amphidinium* isolates based on LM and SEM micrographs are presented for each of the cultivated strains with provisional species assignments: AcSAV105 (*A. carterae*), AcSQ172, AcSQ177 and AcSQ181 (*A.* cf. *carterae*), AmLT112 (*A. massartii*), AA60 (*A. operculatum*), and AtLPZ38 (*A. theodorei*).

### 3.1.1. *Amphidinium carterae* (AcSAV105)

The cells were ovoid to subglobular and varied in length from 9.5 to 12.4 μm (11.1 ± 0.9 μm SD, *n* = 30). This length dimension was significantly shorter in AcSAV105 than for cells of all other isolates (Kruskal–Wallis; *n* = 209; df = 11; *H* = 177.5; df = 6; *p*-value < 0.0001). The cell widths ranged from 5.9 to 8.7 μm (7.7 ± 0.8 μm SD, *n* = 30), significantly narrower than those of isolates AA60, AeSQ177, AeSQ181, and AtLPZ38 (*p*-value < 0.05) (Figure S1). The epicone was small relative to the cell size and formed a crescent upward and to the left (Figure 2a–e).

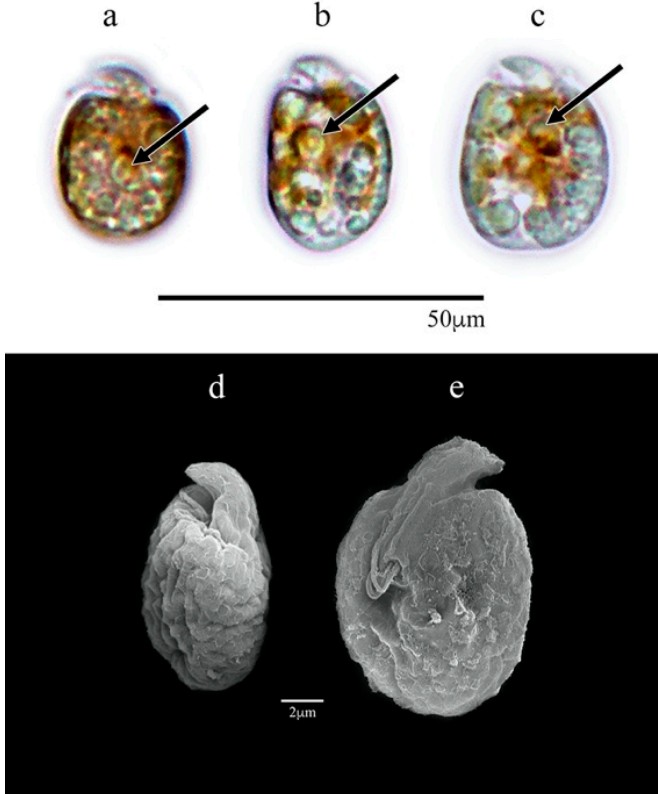

**Figure 2.** *Amphidinium carterae* AcSAV105: (**a**–**c**) cells in LM; (**d**,**e**) cells in SEM; (**a**–**d**) dorsal view, (**e**) ventral view. Pyrenoids are shown with black arrows.

A yellow-brown lobed chloroplast was distributed from the central part to the cell periphery with a central pyrenoid located at the median line or slightly above or behind it (Figure 2a–c). Reddish bodies that could be food particles were present. The nucleus was located at the cell antapex. Flagellar motion, metabolic movement of the pellicle, and amoeboid movement resulting in different cell shapes were frequently observed in active cells.

### 3.1.2. *Amphidinium* cf. *carterae* (AeSQ172, AeSQ177, and AeSQ181)

The cells varied from ovoid to elongated (Figure 3a–h) and varied significantly in cell dimensions among the isolates (Kruskal–Wallis; *n* = 209; df = 6; *H* = 149.58; df = 6; *p*-value < 0.0001; *n* = 30 cells for each isolate) (Figure S1). For AeSQ172, the cell lengths ranged from 10.9 to 15.1 μm (13.0 ± 1.1 μm SD); the widths ranged from 6.4 to 9.7 μm (8.0 ± 0.8 μm SD). For AeSQ177, the cell lengths varied from 13.7 to 16.0 μm (14.7 ± 4.5 μm); the widths varied from 9.1 to 9.9 μm (9.4 ± 2.9 SD). For AeSQ181, the cell lengths ranged from 11.8 to 16.0 μm (13.6 ± 1.1 μm SD); the widths ranged from 6.6 to 10.5 (8.8 ± 0.9 μm SD). In AeSQ172, the epicone was button-shaped, whereas in AeSQ177 and AeSQ181, it was in crescent form.

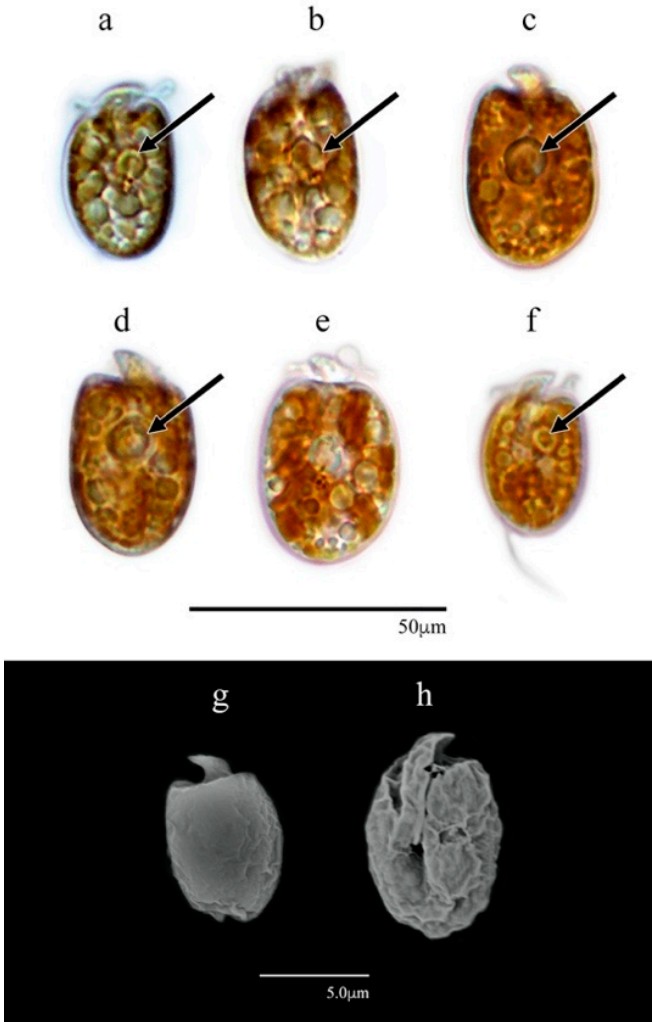

**Figure 3.** *Amphidinium* cf. *carterae* AeSQ172 (**a**,**b**), AeSQ177 (**c**–**e**), and AeSQ181 (**f**–**h**). (**a**–**f**) Cells in LM; (**g**,**h**) cells in SEM; (**a**,**b**,**f**,**h**) ventral view; (**c**,**d**,**e**,**g**) dorsal view. Pyrenoids are shown with black arrows.

A yellow-brown multi-lobed chloroplast was distributed from the central part to the cell periphery. A central or subcentral pyrenoid was located at the median line or slightly above it (Figure 3a–d,f) but was smaller in AeSQ181 (Figure 3f). The nucleus was barely distinguishable but located at the antapex.

The isolates provisionally identified as *Amphidinium* cf. *carterae* but subsequently assigned to *A. eilatiensis* according to the molecular phylogenetic analysis shared many features with *A. carterae*: the cell size and outline; the presence of a single golden-brown multi-lobed chloroplast; the position of the nucleus near the antapex; and the sulcus closer to the right margin of the cell (Figure 3e). The species *A. eilatiensis* is morphologically closest to *A. carterae* and *A. rhynchocephalum*, but it does not have obvious thecal plates (or polygonal units) described for the former species [41]. In the original description of *A. eilatiensis* [41], the sulcus extends about a third of the body length toward the antapex, whereas *A. carterae* is illustrated with the sulcus extending to the antapex [42]. A ventral ridge has been described for both species, but Hulburt [42] did not mention its presence in the original description of *A. carterae*. A more detailed comparison of these features between the two species from Mexico was not feasible. Neither the ventral ridge on the cell surface along the sulcus nor the dorsal position of the chloroplast described for *A. eilatiensis*—two features not included in the original description of *A. carterae* [42]—were observed in the specimens from Mexico.

### 3.1.3. *Amphidinium massartii* (AmLT112)

The cells were apple-shaped, round, oval, or elongated (Figure 4a–e). The cell lengths ranged from 10.6 to 14.6 μm (12.5 ± 1.1 μm SD) and were significantly different from those of all other isolates except for *Amphidinium* cf. *carterae* AeSQ172 ($p$-value > 0.05). The widths varied from 6.5 to 10.4 μm (8.2 ± 1.1 μm SD) ($n = 30$) and showed significant differences from the cells of *A. carterae* AA60 and *A. theodorei* AtLPZ38 ($p$-value < 0.001) (Figure S1). The epicone was small relative to the cell size and formed a crescent upward and to the left. The right margin of the hypocone was more convex than its left margin (Figure 4a,b,e).

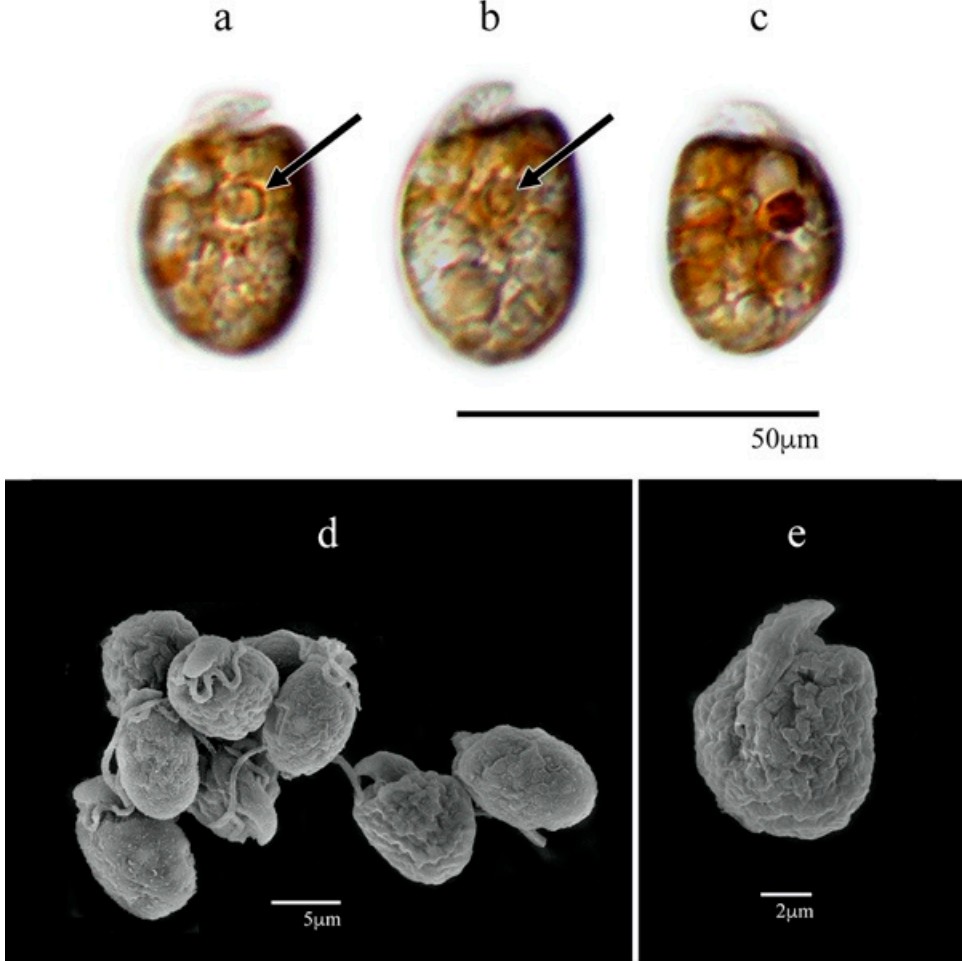

**Figure 4.** *Amphidinium massartii* AmLT112: (**a**–**c**) cells in LM; (**d**,**e**) cells in SEM; (**a**,**b**,**e**) ventral view; (**c**) dorsal view; (**d**) aggregated cells in different views. Pyrenoids are shown with black arrows.

Several yellow-brown elongated chloroplasts were located around the hypocone, and a discoidal pyrenoid was subcentrally located within the cell (Figure 4a,b). The nucleus was likely located at the cell antapex but hardly distinguished.

### 3.1.4. *Amphidinium operculatum* (AA60)

The cells were ellipsoid and almost symmetrical with respect to the longitudinal axis (unlike those of other *Amphidinium* species in this study). For AA60, the cells were widest at the median line or slightly behind it (Figure 5a–d). The cell lengths ranged from 24.0 to 33.0 μm (29.1 ± 2.8 μm SD); the widths ranged from 16.0 to 22.0 μm (19.4 ± 1.7 μm SD) ($n = 30$). The cells of AA60 were significantly larger and greater in both length and width in relation to those of other *Amphidinium* species isolates in this study ($p$-value < 0.001) (Figure S1). The epicone was centered at the anterior end and extended to the left but was poorly distinguished in living cells (Figure 5a,b).

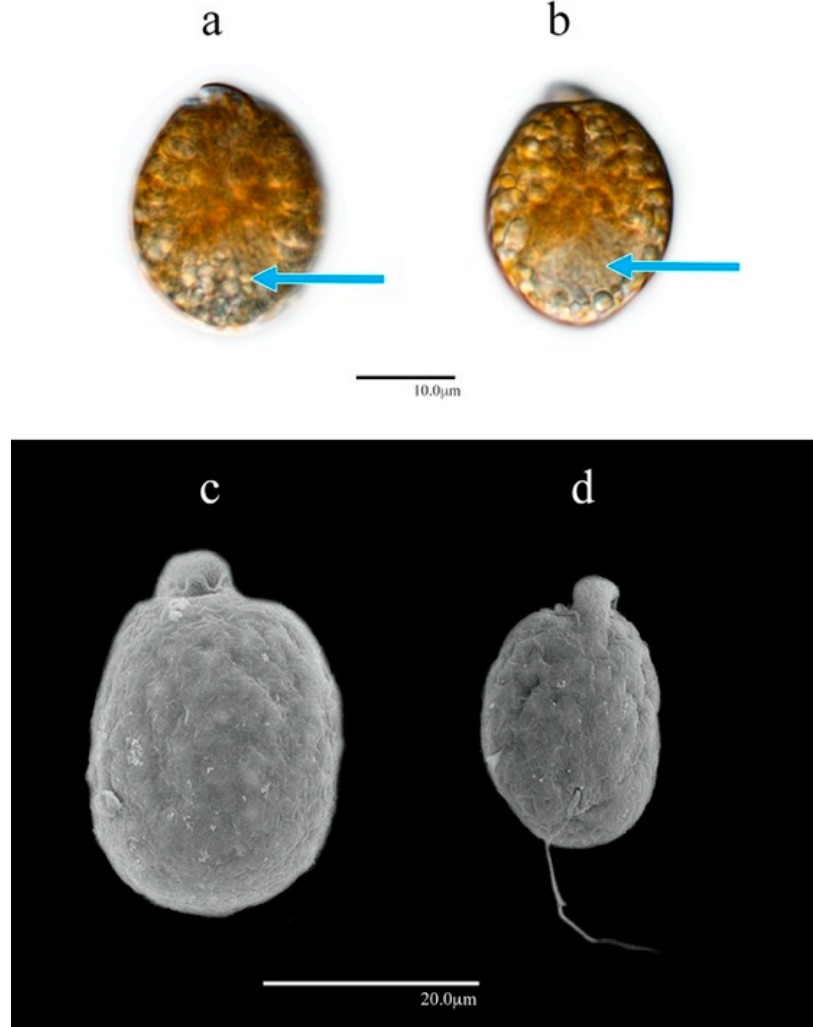

**Figure 5.** *Amphidinium operculatum* AA60: (**a**,**b**) cells in LM in dorsal view; (**c**,**d**) cells in SEM in ventral view. Nuclei are shown with blue arrows.

Multiple orange-yellowish chloroplasts were located in the central zone, and a pyrenoid was barely distinguishable near the median line of the cell. The nucleus was located at the posterior end of the cell (Figure 5a,b).

### 3.1.5. *Amphidinium theodorei* (AtLPZ38)

The cells were of varied shapes from ovoid to subglobular when freshly isolated (Figure 6a–g). The cell lengths varied from 17.8 to 28.1 μm (24.5 ± 2.3 μm SD); the widths ranged from 13.0 to 22.1 μm (18.0 ± 2.0 μm SD) (*n* = 30). These cell dimensions were most similar to those of *A. carterae* AA60 but with significantly smaller cells. Except for AA60, the cells of *A. theodorei* AtLPZ38 were significantly greater in both length and width than those of all other isolates in this study (*p*-value < 0.001) (Figure S1). The epicone was centered and elongated toward the dorsal part of the apex with the button shape well distinguished in living cells (Figure 6a–d), whereas the hypocone usually narrowed slightly toward the cingulum with the widest part closer to the antapex (Figure 6a,b,e,f).

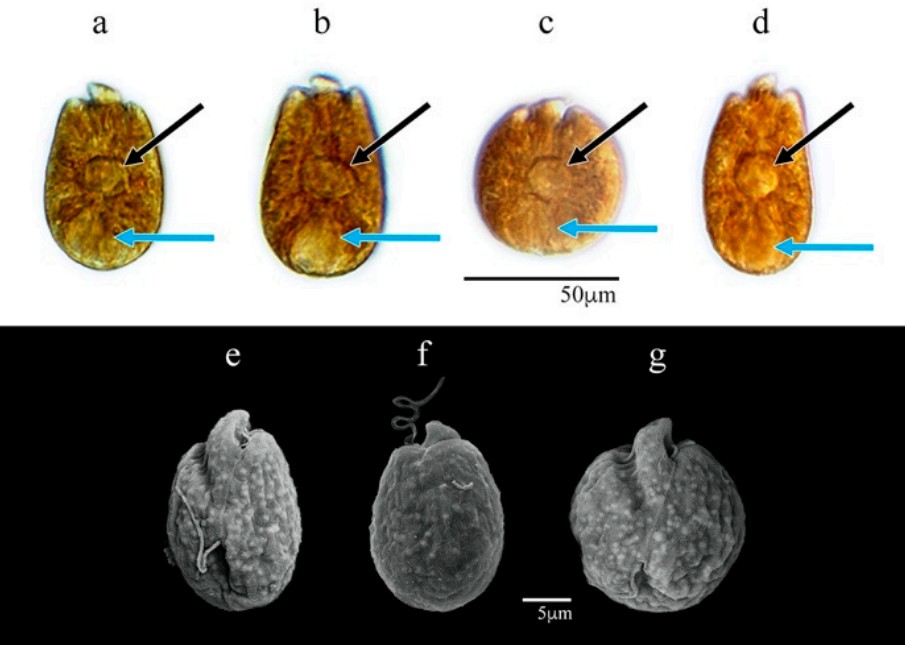

**Figure 6.** *Amphidinium theodorei* AtLPZ38: (**a–d**) cells in LM (bright field); (**e–g**) cells in SEM; (**a**,**b**,**f**) dorsal view; (**c–e**,**g**)—ventral view. Pyrenoids are shown with black arrows and nuclei with blue arrows.

A lobular orange-yellowish chloroplast extended toward the cell periphery in a radial fashion. A relatively large discoidal pyrenoid was centrally located within the cell (Figure 6a–d). Reddish bodies that may have been food particles were observed inside some cells. The nucleus was located at the posterior end of the cell (Figure 6a–d).

### 3.2. Molecular Characterization

Each gene region analyzed showed a particular pattern of nucleotide substitution. The Tamura–Nei model [43] was more suitable for the LSU region, whereas the Kimura two-parameter model [44] was better for the ITS1-5.8S rRNA-ITS2 region. The phylogenetic trees showed different topologies for the two regions analyzed. In fact, analysis of the ITS1-5.8S rRNA-ITS2 region by itself failed to resolve the relationships between isolates AA60, AmLT112, AeSQ181, AeSQ177, and AeSQ172, placing all of them on the same branch. On the other hand, analysis of the LSU region clustered the sequences from isolates AeSQ181, AeSQ177, and AeSQ172 on the same branch as *A. eilatiensis* and *A. carterae* sequences, but none of these were assigned as sister taxa of the isolates from Mexico (Supplementary Materials Figures S2–S5).

These issues were resolved by conducting an analysis of the concatenated gene regions. The phylogenetic analyses of the concatenated gene regions (ITS1-5.8S rRNA-ITS2 + LSU) involved 20 *Amphidinium* species (plus the outgroup sequence of the distantly related dinoflagellate *Heterocapsa* sp.) (Table 2) for a total of 2160 nucleotide positions analyzed. The concatenated sequences displayed a different pattern of nucleotide substitution represented by the General Time-Reversible model. The tree topologies from both ML and MP approaches were congruent with some exceptions: the ML tree failed to assign a sister taxon to isolate AmLT112 (Figure 7), while the MP tree clustered it with *A. massartii* (Figure 8). Although the ML tree was suitable for representing genetic distances as branch lengths, the tree topology from the MP method was more congruent with the morphological observations described before in the present study: the sister taxa assigned for our isolates in the consensus MP tree coincided in all cases with species assignments via morphological observations.

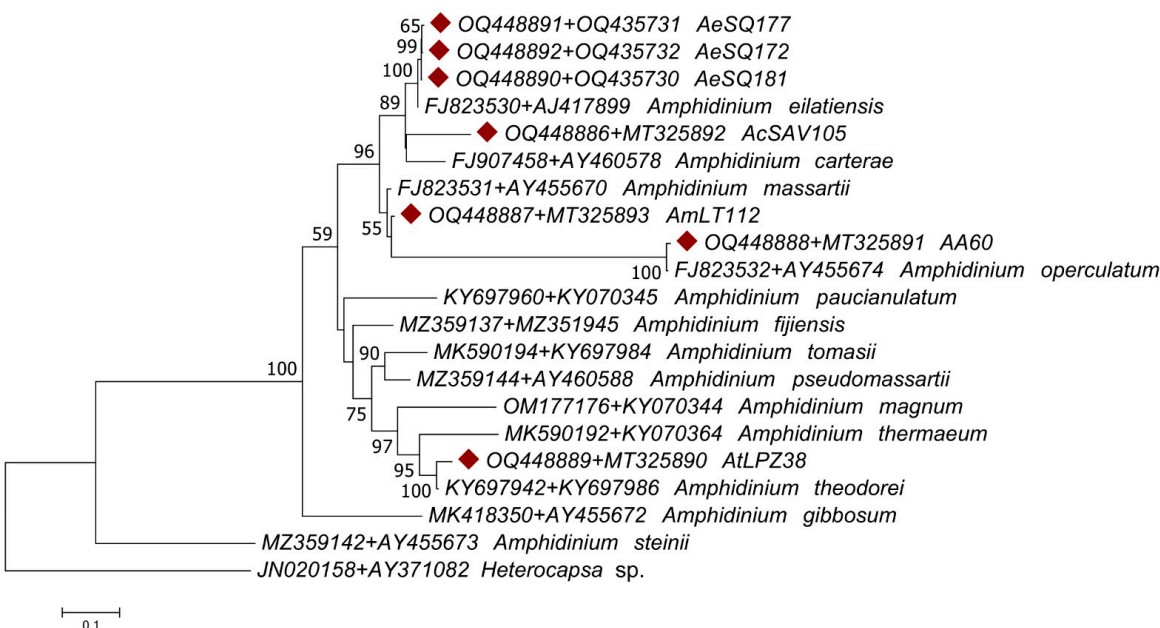

**Figure 7.** Maximum likelihood (ML) phylogenetic tree of *Amphidinium* taxa based on ITS1-5.8S rRNA-ITS2 + LSU concatenated sequences, including the isolates sequenced in the present study (diamond shapes). Branch lengths represent the genetic distances among the analyzed sequences, and numbers next to the nodes represent the clustering support after 2000 bootstrap replicates; only values above 50% are displayed. Sequence labels include the accession numbers of the concatenated gene sequences and the species name when assigned.

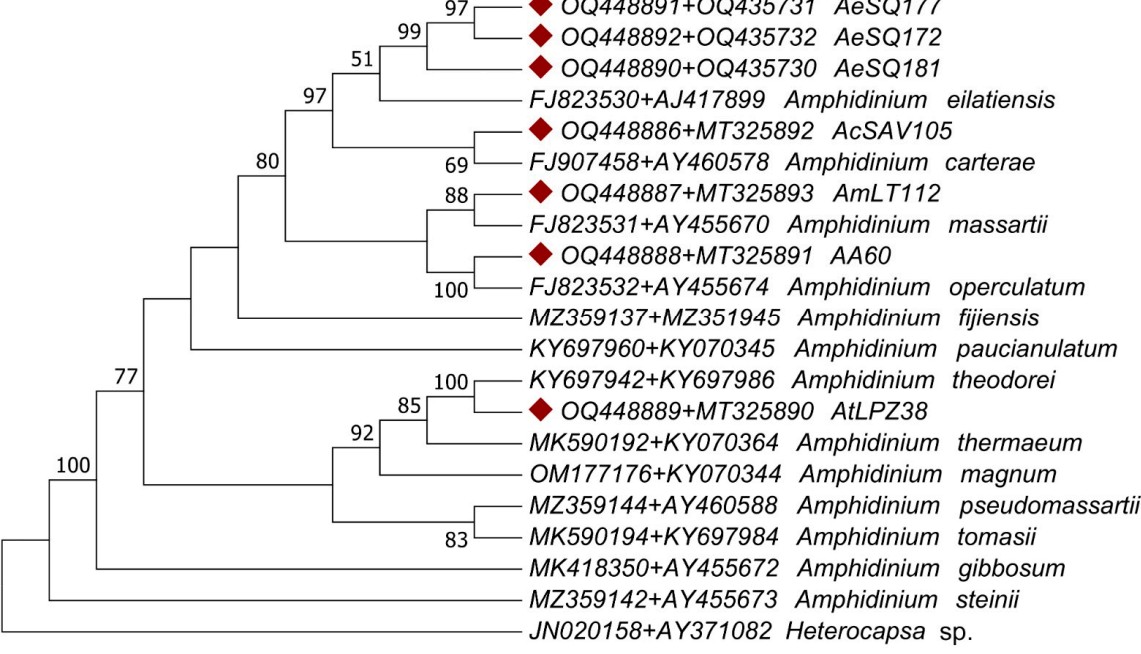

**Figure 8.** Maximum parsimony (MP) phylogenetic tree of *Amphidinium* taxa based on ITS1-5.8S rRNA-ITS2 + LSU concatenated sequences, including the strains sequenced in the present study (diamond shapes). The consensus tree topology is presented; the numbers next to the nodes represent the clustering support after 2000 bootstrap replicates. Only values above 50% are displayed. Sequence labels include the accession numbers of the concatenated gene sequences and the species name when assigned.

The consensus MP phylogenic tree grouped all isolates from Mexico within the Operculatum clade *sensu lato* [1]. As expected from the preliminary morphological analysis, the ITS1-5.8S rRNA-ITS2 + LSU concatenated sequence of AcSAV105 from Veracruz grouped with *A. carterae*; the sequence of AmLT112 from Campeche grouped with *A. massartii*; and that of AA60 from Veracruz grouped with *A. operculatum*. The AtLPZ38 sequence from Baja California Sur was clustered with *A. theodorei*. The isolates AA60 and AtLPZ38 were clustered with their sister taxa with 100% bootstrap support on both the MP and ML trees. Sequences from AeSQ-172, AeSQ-177, and AeSQ-181 from Baja California provisionally identified via morphological criteria as *A.* cf. *carterae* (Table 1) clustered with *A. eilatiensis* in the MP phylogenetic reconstruction (Figure 8). This constituted the first confirmed report of this latter species from Mexico or indeed from anywhere else in the Americas.

## 4. Discussion

The genus *Amphidinium* exhibits a widespread geographical distribution from the tropics to high-latitude waters and is therefore considered cosmopolitan [1]. Until now, there have been few records of the phylogenetic relationships of *Amphidinium* species from Mexico, and identifications have not usually been supported by molecular sequencing data. Defining the taxonomy of *Amphidinium* species solely based on morphological characters is complicated, in part because their morphology varies according to the strain and the life stage in which they are observed [1,4,21,34]. This high degree of morphological plasticity and the cryptic biogeographical distribution in diverse marine habitats has almost certainly led to underestimation and misidentifications in routine field surveys of benthic and pelagic microeukaryote communities. Delicate athecate dinoflagellates such as *Amphidinium* also fail to preserve well in conventional fixatives for later morphological identification.

Many toxigenic bHAB species (including those of *Amphidinium*) have undergone an apparent expansion of geographical distribution in Latin America in recent years [8]. The lack of confirmed bHAB events associated with *Amphidinium* blooms in Mexico has tended to cause neglect of this genus as a high priority for regional toxin-monitoring programs. Members of the genus *Amphidinium* may be occupying new ecological niches due to coastal eutrophication, unintentional introduction of invasive species, and climate change, but this cannot be substantiated without more intensive efforts to determine biogeographical distribution from field surveys and critical taxonomy.

### 4.1. Morphological Characterization

The high similarity between closely related species and morphological plasticity are factors that complicate morphological identification. The apparent overlap and variation in some features such as the cell size and shape mean that *Amphidinium* species are often poorly characterized, especially when observed only under LM. With SEM, micro-structural details of the cell surface can be resolved for more reliable species identification. In the cases of *A. carterae* and *A. massartii* [34] as defined herein, these species differed primarily in the size and shape of the lobulated chloroplast, which was distributed toward the cell periphery in *A. carterae* but more dispersed in *A. massartii* [45]. The species also differed in the patterning of the external membrane and the presence of scales in *A. massartii* [1]. Unfortunately, without observation under transmission electron microscopy (TEM), scales located outside the plasma membrane could not be distinguished. In scale-bearing *Amphidinium* species, scales were not visible via SEM because they were tiny; e.g., only 136.4 × 91.0 × 81.8 nm in *A. cupulatisquama* Tamura & Horiguchi [46], 64.8–68.4 × 48.5–49.3 × 12.6–18.4 nm in two unidentified *Amphidinium* strains closely related to *A. carterae* [47], and 276 ± 17 nm in *A. stirisquamtum* Z. Luo, Na Wang & H. Gu [48].

The flagellar insertion point was slightly lower in *A. massartii* than in *A. carterae* [40], and the epicone was narrower in *A. massartii* than in *A. carterae* [22]. Considering the clonal variation in the aforementioned characteristics, the morphological features of the cultured isolates of *A. carterae* and *A. massartii* coincided well with those reported in the literature.

In specimens provisionally assigned to *A.* cf. *carterae* based on preliminary LM observations (*A.* cf. *carterae* AeSQ172, AeSQ177, and AeSQ181), we could not distinguish the lobes of the chloroplast. In the original description of *A. eilatiensis* [41], six lobes were indicated, but the observations were obtained by confocal microscopy, which provides superior resolution and depth of field. The authors [41] emphasized that unlike *A. carterae*, *A. eilatiensis* "does not have the obvious thecal plates or polygonal units". In our specimens, we also did not observe the ventral ridge—a distinctive feature of this species—located between the two flagellar pores.

The morphology of *A. operculatum* AA60 coincided with descriptions in the literature [4]. This species was especially distinguished by the position of the flagellar insertion located at the antapex of the cell and away from the epicone (Figure 5). This feature was in contrast to *A. carterae* and *A. massartii*, for which the flagellar insertion point was located closer to the epicone (Figures 2 and 4). Cultured *A. operculatum* cells were the largest among all isolates of *Amphidinium* species (see Section 3.1.4 and Supplementary Materials Figure S1) evaluated in this study.

The high degree of morphological plasticity in *Amphidinium*—depending upon environmental factors and life stages—was corroborated with *A. theodorei* (AtLPZ38). When this strain was freshly isolated, its cell shape could be either round or oval (Figure 6). After a few weeks in culture, the formation of cysts surrounded by mucilage was also noticed. The morphology of AtLPZ38 was somewhat different from the Fiji strain Amth0702-1, although both shared the epicone shape and cyst formation feature [1]. Unlike the illustrated cells of Amth0702-1 [1], AtLPZ38 cells differed as follows: (1) the sides of the cell could be symmetrical in the ventral or dorsal view in relation to the longitudinal axis (Figure 6a–c,f,g); (2) the pyrenoid was always central (Figure 6a–d); (3) the maximal cell width could be behind the median line of the cell (Figure 6a,b); and (4) the insertion point of the longitudinal flagellum was also located behind the median line (Figure 6e). The molecular sequence data herein (Figures 7 and 8) indicated that they belonged to the same species. Nevertheless, it would be informative to compare the differences in the time frame of the photographic records over the respective culture cycles to confirm the degree of cellular plasticity and morphological variation between the strains from Mexico and Fiji.

Reddish bodies such as those observed in the cells of *A. carterae* (AcSAV105) (Figure 2) and *A. theodorei* (AtLPZ38) (Figure 6) have been frequently reported in cells of other dinoflagellate genera such as *Coolia* Meunier and *Ostreopsis* Schmidt. These reddish bodies have been attributed to mixotrophic activity of these dinoflagellates, although this has not been proven [49]. These reddish bodies usually disappear within several cell division cycles after the dinoflagellates are maintained unialgally in defined inorganic seawater-based culture medium as evidence of lack of phagotrophy.

### 4.2. Molecular Phylogenetic Analysis

Compared to other groups of dinoflagellates, the genus *Amphidinium* comprises high interspecific genetic diversity and intraspecific variability [1,45]. Molecular analyses of the D1-D6 domains of the LSU rRNA gene have indicated high variability in these regions, suggesting either that infrageneric taxa have a high rate of evolution in their respective rRNA genes or that they exhibit greater species diversity because they are older than other groups [4,45,50]. These alternative scenarios are not mutually exclusive, but the issue has not been resolved.

The analyzed gene regions showed different patterns of nucleotide substitution, suggesting that each region was subject to different mutation rates and evolutionary dynamics. The sequenced gene regions yielded different tree topologies with the patterns of ancestral branches differing between the ML and MP methods. Although the ML tree was suitable for representing genetic distances as branch lengths, the tree topology from the MP method was more congruent with the morphological observations of species described before and in the present study. The unified phylogenetic tree presented herein—the first based on concatenated sequences from different genes for the genus *Amphidinium*—clearly improved

the topologies obtained for each gene separately. Specifically, the trees obtained from the concatenated dataset (Figures 7 and 8) solved the problems in the phylogenetic reconstruction based solely on individual gene regions (Figures S2–S5). For example, the phylogeny of the LSU region by itself also failed to resolve the morphologically complex relationships between AeSQ181, AeSQ177, AeSQ172, *A. eilatiensis*, and *A. carterae* (Supplementary Materials Figures S2 and S3). The phylogeny of the ITS1-5.8S rRNA-ITS2 region failed to assign sister taxa to the isolates AA60, AtLPZ38, AeSQ181, AeSQ177, and AeSQ172, clustering them all on the same branch (Supplementary Materials Figures S4 and S5). By contrast, the consensus MP phylogeny for the concatenated gene regions clustered all new isolates from Mexico with their expected sister taxa according to the previous species assignments. Preliminary observations of morphological characteristics from new isolates AeSQ181, AeSQ177, and AeSQ172 led to a vague assignment to *A.* cf. *carterae*, but the molecular phylogeny persuasively argued for a closer affinity with *A. eilatiensis.*

The fact that the consensus MP method yielded phylogenetic reconstruction more congruent with the morphological observations than the ML method may be related to differences in mutation rates between the gene regions. The reliability of the ML method depends on an accurate mutation model; combining genes with different mutation rates could therefore affect the reliability of this method [51]. By contrast, the MP method reconciles the differences in the mutation dynamics between both gene regions since it assumes a unique phylogenetic criterion in which the shortest possible tree that explains the data is considered the best without assuming a particular mutation model [51]. In this regard, the MP approach has been shown to be a robust model for reconstructing dinoflagellate evolutionary pathways at various levels [2,52]. We point out that the primary purpose of our phylogenetic analysis was to provide support for accurate species identification and assignment in combination with the morphological analyses. We detected major differences in the ancestral nodes between ML and MP trees; our intention, however, was not to attempt a phylogenetic revision of the Amphidiniales or even the diversity within genus *Amphidinium* with such a restricted new sequence data set from a few isolates and limited archived sequences.

In this regard, previous molecular phylogenetic studies on the *Amphidinium* genus have proposed the existence of two major clades: the Operculatum and the Herdmanii clades [1,34,40]. These clades show strong differences in their evolutionary rates since the Operculatum clade includes the vast majority of the *Amphidinium* species, thereby indicating a higher diversification rate than for the Herdmanii clade. These major clades were also distinguished in our phylogenetic analysis, in which the most ancestral branch in the phylogenetic tree separated the only taxon of the Herdmanii clade (*A. steinii* (Lemmerm.) Kof. & Swezy) from the rest of the species [1,34,40]. Our results suggested that *A. gibbosum* represents the basal branch of the Operculatum clade instead of *A. operculatum*, which was placed in a more terminal position than previously reported [1,34,40]. For a better interpretation of deeper evolutionary pathways within the *Amphidinium* genus, we recommend developing new DNA barcodes (including genomic regions with lower mutation rates), especially to solve the ancestral nodes, which was outside of the scope of the present study. We consider that the new trees presented herein are most useful for species identification.

Molecular phylogenies of *Amphidinium* also concur in the relatedness between the species *A. carterae* and *A. eilatiensis* [40], which was also reflected in our MP phylogenetic tree (Figure 8). The degree of genetic relatedness found between the *A. carterae* strain (AcSAV105) isolated from the Gulf of Mexico and the putative *A. eilatiensis* strains (AeSQ-172, AeSQ-177, and AeSQ-181) from the geographically disjunct Gulf of California suggests vicariance between these species as opposed to dispersal mixing of *Amphidinium* genotypes. Such vicariance could have occurred after the emergence of the Panama isthmus, impeding gene flow between the Atlantic and Pacific ocean basins [53]. If this hypothesis is confirmed with further research, such vicariant processes could be the basis for estimating mutation rates and the calibration of molecular clocks [54] for *Amphidinium* species.

In conclusion, this study of morphological and molecular characteristics of the benthic dinoflagellate *Amphidinium* from selected coastal waters of Mexico provides new insights into the regional biography of members of this genus. The revised molecular phylogenetic analysis was consistent with and did not challenge the traditional subdivision of the genus *Amphidinium* into two sister clades but confirmed the first records of *A. theodorei* and *A. massartii* from coastal waters of Mexico. More significantly, the molecular phylogenetic evidence indicated that *Amphidinium* populations heretofore described as *A.* cf. *carterae* from Baja California may in fact belong to *A. eilatiensis* [41]. High morphological plasticity within the genus revealed cells with similar dimensions and marginal outlines in all species but one (*A. operculatum*) as well as similar positions of the nucleus and pyrenoid. In several cases, the infraspecific morphological variation among isolates within the defined *Amphidinium* species was greater than the interspecific variation.

Molecular data based on ITS1-5.8S rRNA-ITS2 + LSU concatenated sequences were proven to be essential to identify and distinguish *Amphidinium* species. The case for transferring *A.* cf. *carterae* from Baja California to *A. eilatiensis* may be supported by retrospective analysis of published studies from the region [11,12]. Further support for these provisional species assignments is contingent upon a more detailed analysis of ITS2 secondary structure not currently available from a larger pool of closely related isolates. Future biogeographical investigations of *Amphidinium* in the Gulf of California by combining morphological and molecular techniques will help to unravel biogeographical distribution patterns and species affiliations. This distinction is of more than just taxonomic and nomenclatural significance. *Amphidinium eilatiensis* was first described from a bloom in a mariculture sedimentation pond near the Gulf of Eilat, Israel [41], but has never been implicated in any global HAB or toxigenic events. *A. carterae* has long been recognized to produce a suite of bioactive (some cytotoxic) amphidinols (AM) and was circumstantially linked to fish-killing incidents in a few cases. Just recently, putative *A. eilatiensis* isolates AeSQ172, 177, and 181 from the northern Mexican Pacific coast were confirmed to produce AMs in higher quantities and varieties than any global *Amphidinium* isolates known to date [55]. Resolution of the complex nature of species distinctions and production of potentially toxic secondary metabolites—including AM analogs—is therefore key to defining the emerging but undefined risk of *Amphidinium* blooms to human health and seafood security on the Pacific coast, the Gulf of California, and the southern Gulf of Mexico.

**Supplementary Materials:** The following supporting information can be downloaded at: https://www.mdpi.com/article/10.3390/phycology3020020/s1. Figure S1: Comparison of morphometric characteristics of cultured *Amphidinium* species from Mexican coastal waters (*n* = 30 for each isolate). *A. carterae* = AcSAV105; *A.* cf. *carterae* = AeSQ172, AeSQ177, and AeSQ181; *A. massartii* = AmLT112; *A. operculatum* = AA60; *A. theodorei* = AtLPZ38. The line inside the boxes indicates the median, the whiskers represent the highest and lowest values excluding outliers, and the transparent points represent the underlying distribution of the data. Different letters (a,b,c,d) above the boxes indicate significant differences at 0.05% error. Figure S2: Maximum likelihood tree for large ribosomal subunit (LSU) gene sequences of *Amphidinium* species. Sequences obtained in the present study are marked with diamond shapes. The percentage of trees in which the associated taxa clustered together (after 2000 bootstrap replicates) is shown next to the branches. A discrete Gamma distribution was used to model evolutionary rate differences among sites (+G; parameter = 0.8832). Sequence labels include the strain name, GenBank accession number, and species name. Figure S3: Maximum parsimony consensus tree for large ribosomal subunit (LSU) gene sequences of *Amphidinium* species. Sequences obtained in the present study are marked with diamond shapes. The percentage of trees in which the associated taxa clustered together (after 2000 bootstrap replicates) is shown next to the branches. Sequence labels include the strain name, GenBank accession number, and species name. Figure S4: Maximum likelihood tree for the ITS1-5.8 S rRNA-ITS2 region of *Amphidinium* species. Sequences obtained in the present study are marked with diamond shapes. The percentage of trees in which the associated taxa clustered together (after 2000 bootstrap replicates) is shown next to the branches. A discrete Gamma distribution was used to model evolutionary rate differences among sites (+G; parameter = 2.1612). Sequence labels include the GenBank accession

number, species name, and strain name. Figure S5: Maximum parsimony consensus tree for the ITS1-5.8 S rRNA-ITS2 region of *Amphidinium* species. Sequences obtained in the present study are marked with diamond shapes. The percentage of trees in which the associated taxa clustered together (after 2000 bootstrap replicates) is shown next to the branches. Sequence labels include the GenBank accession number, species name, and strain name.

**Author Contributions:** L.M.D.-R.: Conceptualization; field sampling; *Amphidinium* cell isolation, culture, and harvest; provision of light micrographs; leading and coordinating the drafting of the manuscript; supervision of research team and data analysis. O.E.J.: Formal bioinformatic data analysis; molecular data curation and visualization; manuscript writing, reviewing, and editing. Y.B.O.: Morphological and taxonomic analysis and manuscript writing. A.L.M.-C.: Cell isolation, culture, and harvest; molecular sequencing; and analysis under F.R.-C.'s and L.M.D.-R.'s supervision. F.R.-C.: DNA extraction, molecular analysis and supervision, and manuscript writing. D.C.-G. and M.d.C.O.-R.: *Amphidinium* cell culture and harvest, DNA extraction and amplification, molecular data analysis under L.M.D.-R.'s and O.E.J.'s supervision. V.A.C.-U.: Statistical data analyses and figure plotting. A.D.C.: Conceptualization and critical data analysis, field sampling for isolation; coleading and coordinating drafting of the manuscript with a focus on interpretation of the results and discussion. All authors have read and agreed to the published version of the manuscript.

**Funding:** This research was funded primarily by the Consejo Nacional de Ciencia y Tecnología (CONACyT, Mexico) through the Basic Science Project (number A1-S-8616) and the Cátedra CONA-CyT (Investigadoras e Investigadores por México, CONAHCYT) Project 1009 (both awarded to L.M.D.-R.). The Helmholtz research program "Changing Earth, Sustaining our Future" (Subtopic 6.2—Adaptation of marine life) of the Alfred-Wegener-Institut, Helmholtz Zentrum für Polar- und Meeresforschung, Germany, supported the participation of A.D.C. and L.M.D.-R. and contributed to the publication of this work.

**Institutional Review Board Statement:** Not applicable.

**Informed Consent Statement:** Not applicable.

**Data Availability Statement:** The alignments presented in this study can be made available upon request. Data are available at https://www.ncbi.nlm.nih.gov/genbank accessed on 3 March 2023 (ITS1-5.8S rRNA-ITS2 GenBank accession numbers: OQ448886—OQ448892; LSU rRNA GenBank accession numbers: MT325890—MT325893 and OQ435730—OQ435732).

**Acknowledgments:** We thank Manuel Victoria-Muguira of the *Dorado Buceo* for boat access and logistical support of field sampling from Veracruz and Manuel Rodríguez-Gómez of the *Acuario de Veracruz* for donating filtered seawater. Mario Fernando Sánchez-Bernal was responsible for imaging and curation. We gratefully acknowledge Edna Sánchez-Castrejón from the Genomics Laboratory at CICESE for her competent support in the sequencing laboratory.

**Conflicts of Interest:** The authors declare no conflict of interest. The funders had no role in the design of the study; in the collection, analyses, or interpretation of data; in the writing of the manuscript; or in the decision to publish the results.

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
