# Peer review of "Morphological and Molecular Characterization of the Benthic Dinoflagellate Amphidinium from Coastal Waters of Mexico"

_phycology, doi:10.3390/phycology3020020_

Round 1

Reviewer 1 Report

This study contributes to genotyping of Amphidinium samples from diverse coastal locations on the NE Pacific, Gulf of California, and southern Gulf of Mexico. There are critical morphological analysis by photonic and scanning electron microscopy, followed by phylogenetic reconstruction based on nuclear-encoded, partial, large subunit (LSU) rDNA and internal transcribed spacers I and II (ITS1 and ITS2) sequences. Despite an interesting idea, and good microscopic parts (LM, SEM images and their interpretation), the half of the manuscript represents the results and discussion of molecular phylogenetic analyses with dubious species identification. So, I have to redirect the article to the major revision.

Please, see my comments below:

1.      I highly doubt the need for such keywords: amphidinolide; benthic Harmful Algal Bloom (bHAB); polyketide toxin. The article does not directly address these aspects. I propose to add a few more relevant keywords.

2.     It is better to move Fig.1 to page 4, close to reference to Fig.1.

3.    As a political map presented, please add countries names (maybe main only).

4.    The name of Table 2 is not appropriate. It needs to be «Table 2. Taxa, strain ID numbers, Genbank accession numbers, and references of all Amphidinium species used in this study.» The paragraph (Lines 272-278), please put into M&M.

5.      Please delete «.1» in all cases after GenBank Acc. # through the text.

6.      Please replace «ITS1-5.8S_rRNA-ITS2» to «ITS1-5.8S rRNA-ITS2» through the text (for example, line 273).

7.      The sentence «The selected tree represents the evolutionary history of the analyzed taxa performed with MEGA7» (L 297-298) is quite controversial, so I propose to delete it.

8.      Please, italicize the species Latin names at Lines 305, 321, 353, 367, and 381.

9.      The caption of the Fig.6 please try to replace page 12.

10.  The main cause for concern is phylogenetic tree. I didn’t see the congruence between ML and MP analyses, while only one MP tree presented, and supports also attributed to MP only. Overall, I was under the impression that merging the LSU and ITS rRNA region datasets was not the best idea. The mutational dynamics of these two regions is not similar and in the paper you could compare single region phylogeny first, and concatenated results next. The tree contains too small sampling probably due to comparisons between the same organism DNA fragments. I’m not sure that every representative taxa from other studies are holotypes of the corresponding species, please check. To my mind the identification is weak in the cases of A. theodorei, A. carterae, and A. operculatum. In this cases genetic distances on the tree are seem to be interspecific, not intraspecific. It is probably a new species to science. Neither morphology, nor this simple phylogeny could you help here. I suppose in this case you need to check the secondary structures of the ITS2 from the Karafas et al., 2017. So, you could prove your statements. If possible, add p-distance analyses, sec. structures, and more representative phylogeny LSU, ITS, probably ITS2 or concatenated, but with strong argumentation.

11.  Please, add your GB numbers on the tree.

12.  Discussion needs to be revised according to the results of analyses.

13.  I suggest deleting paragraph at lines 505-510.

Nevertheless, the points below speak in favor of accepting the manuscript for the journal:

·         The manuscript is clear, relevant for the field and well-structured.

·         The abstract presented in the article characterizes the subject, reflects the purpose of the study, the main content and novelty of the article.

·         The introduction contains historical and theoretical data according to modern literary sources.

·         The authors used new research methods and computer programs.

Author Response

We have focused most attention on the reviewer’s comments on the doubtful phylogenetic analysis and species assignments.

  1. I highly doubt the need for such keywords: amphidinolide; benthic Harmful Algal Bloom (bHAB); polyketide toxin. The article does not directly address these aspects. I propose to add a few more relevant keywords.

We acknowledge this valid point and have replaced keywords.

  1. It is better to move Fig.1 to page 4, close to reference to Fig.1.

Agreed and moved as suggested

  1. As a political map presented, please add countries names (maybe main only).

Agreed. We added the “Mexico” label because all samples came from Mexican coastal waters but also USA, Guatemala, and Belize labels were added to the map.

  1. The name of Table 2 is not appropriate. It needs to be «Table 2. Taxa, strain ID numbers, Genbank accession numbers, and references of all Amphidinium species used in this study.» The paragraph (Lines 272-278), please put into M&M.

The reviewer is correct, and we have modified the text and table legend exactly as requested.

  1. Please delete «.1» in all cases after GenBank Acc. # through the text.

Changed as requested.

  1. Please replace «ITS1-5.8S_rRNA-ITS2» to «ITS1-5.8S rRNA-ITS2» through the text (for example, line 273).

Changed as requested.

  1. The sentence «The selected tree represents the evolutionary history of the analyzed taxa performed with MEGA7» (L 297-298) is quite controversial, so I propose to delete it.

This is a fair point by the reviewer, and we have complied with the deletion.

  1. Please, italicize the species Latin names at Lines 305, 321, 353, 367, and 381.

Lack of italics for some species names was an artifact of autoformatting conversions and has been corrected.

  1. The caption of the Fig.6 please try to replace page 12.

If the reviewer is referring to placement of the figure or legend on p. 12, this is a formatting decision correctable for publication. Figures and captions will be put together in the last phases of the galley proof presentation and will not be placed on different pages. However, if this should occur, we will ask the editors to change it during the galley proofing.

  1. The main cause for concern is phylogenetic tree. I didn’t see the congruence between ML and MP analyses, while only one MP tree presented, and supports also attributed to MP only.

Thanks for your observation. Since the MP tree showed higher congruence with the morphological observations and previous phylogenetic studies on Amphidinium, we omitted the ML tree in the original manuscript. Following your suggestion, we are including the ML tree in the revised version for comparison, bringing transparency to our arguments. Explanatory text has been added where appropriate to the M&M, Results, and Discussion.  

  1. Overall, I was under the impression that merging the LSU and ITS rRNA region datasets was not the best idea. The mutational dynamics of these two regions is not similar and in the paper you could compare single region phylogeny first, and concatenated results next.

Thank for your constructive suggestion; we are aware of many publications where simple merging the LSU and ITS rRNA datasets is common practice. Nevertheless, we concur with the reviewer that the more conservative approach of sequential comparison is likely better. In the re-analysis we produced the individual single region phylogeny first, and then concatenated results.  We have therefore produced separate phylogenetic trees for each region and compared them. The analytical data for these trees is included as Supplementary Material. A section on the results of this comparison is now included in the Discussion.

  1. The tree contains too small sampling probably due to comparisons between the same organism DNA fragments.

We have produced new phylogenetic trees, incrementing the number of bootstrap replicates from 200 to 2000.

  1. I’m not sure that every representative taxa from other studies are holotypes of the corresponding species, please check. To my mind the identification is weak in the cases of theodorei, A. carterae, and A. operculatum. In this cases genetic distances on the tree are seem to be interspecific, not intraspecific. It is probably a new species to science.

We have done our best to validate the naming of the species assigned to the corresponding sequences downloaded from GenBank, but no doubt there are misassignments or necessary revisions. That the “identification is weak” at the morphological level is more a function of variation and vague species descriptors than our analysis. Furthermore, the reviewer may indeed be correct that, e.g., the cases of isolates assigned to A. eliatiensis from Mexico represent a “new species”.  But we are hesitant to describe them as such because of the morphological similarity to A. cf carterae and the admittedly somewhat ambiguous genetic distances on the original tree. After further phylogenetic analysis based on alternative trees, we are persuaded to stick with our original provisional species assignment of these isolates, since the concatenated tree agrees with the “morphospecies”.

  1. Neither morphology, nor this simple phylogeny could you help here. I suppose in this case you need to check the secondary structures of the ITS2 from the Karafas et al., 2017. So, you could prove your statements. If possible, add p-distance analyses, sec. structures, and more representative phylogeny LSU, ITS, probably ITS2 or concatenated, but with strong argumentation.

Thanks a lot for this helpful observation, which we acknowledge is a potential weak point  of our analysis. Without the detailed analysis of ITS2 secondary structure (CBC) on a lot more isolates (not currently available) our infraspecific arguments on Amphidinium diversity and species boundaries would be difficult to sustain. Such work would only be undertaken to evaluate infraspecific diversity or reorganization of the Amphidiniales – outside the scope of the current manuscript.

We were using highly divergent sequences of A. operculatum, retrieved from GenBank, some of which were possibly misidentified to species in the original assignments (and subsequently revised in more rigorous taxonomic revisions). Of course, this may have affected genetic distance calculations, but we have taken care to weed out these possible taxonomic anomalies. For example, sequences from strain SKLMP_S091 were replaced with those from strain UTEX_1946 (voucher specimen). After replacing these sequences, we ran new phylogenetic trees using both ML and MP approaches. The distance calculation was performed only for the ML tree since it was based on a nucleotide substitution model, as previously estimated. The genetic distances obtained were slightly shorter than those obtained in the original harvest for the submitted manuscript, falling within the expected ranges. However, despite the fact that the ML method assigned our AmLT112 strain to the same branch of A. massartii, it failed to cluster them as sister taxa. On the other hand, the MP method solved this branch complex better, since it clearly clustered AmLT112 with A. massartii as sister taxa. This was congruent with the morphological observations; therefore, we consider that despite the fact that the ML method was perhaps more suitable for calculating genetic distances, the MP method solved better the tree topology. We contend that it is best to present the alternative trees from different phylogenetic approaches. Then, integrating and reconciling the information from alternatives trees, with that obtained from the morphological observations, will offer the most reliable description of phylogenetic and taxonomic relationships.

Finally, we would like to point out that the primary purpose of our phylogenetic analysis was to provide additional support for species identification, in combination with the morphological analyses. This was achieved with the concatenated MP tree, since the tree topology coincided with the species assignation suggested by the morphological characteristics. We did not attempt to offer a definitive phylogenetic reconstruction of the genus Amphidinium with the Amphidiniales or infraspecific molecular diversity. We consider, therefore, that our trees herein are most useful for species assignment only. For a better interpretation of evolutionary pathways within the Amphidinium genus we recommend developing new DNA barcodes (including more genomic regions, especially to solve the ancestral nodes) based on more isolates from geographical populations. That was outside of the scope of the present study, given the paucity of comparative isolates and sequences.

  1. Please, add your GB numbers on the tree.

These are now included

  1. Discussion needs to be revised according to the results of analyses.

We have revised the Discussion in the light of replotting of the sequence data for alternative phylogenetic reconstructions. We also included relevant information on this topic in the M&M, Results and Supplementary Section but much shorter in the Discussion than the long-winded explanations given above in point 14 above.

     I suggest deleting paragraph at lines 505-510.

Agreed, this was general background text but appeared orphaned and not specifically relevant to the phylogenetic analysis as presented in this paper. We did recover and relocate some of the text to better support the more speculative evolutionary interpretations.

 Nevertheless, the points below speak in favor of accepting the manuscript for the journal:

  • The manuscript is clear, relevant for the field and well-structured.
  • The abstract presented in the article characterizes the subject, reflects the purpose of the study, the main content and novelty of the article.
  • The introduction contains historical and theoretical data according to modern literary sources.
  • The authors used new research methods and computer programs.

Thank you. We interpret this as a generally favorable review warranting publication after addressing the phylogenetic tree issues noted above.

Reviewer 2 Report

This work describes Ampidinium species diversity in Mexico, with reporting two first records of A. theodorei and A. massartii. This diversity was evaluated with morphological and molecular analysis. The purpose of the study, the method commonly used, and the results presented are appropriately described in the MS. Thus, it can be published after minor revisions as follows.

Line 26 The dinoflagellate genus Amphidinium -> The genus Amphidinium

Line 37 Herdmanii and Operculatum -> Italic

Line 182 50 mmol -> umol

In result section, It would be better if authors indicate the position of the nuclei in Photo images of each species.

In 3.1.3. Amphidinium massartii (AmLT112), In the discussion, it was explained that there is also the presence or absence of scale when distinguishing A. carterae and A. massartii by their morphological characteristics. Therefore, it would be nice to have data on the presence or absence of scale in the results of this paper.

Line 414 Oper-414 culatum clade sensu lato -> Oper-414 culatum clade stricto

Author Response

This work describes Amphidinium species diversity in Mexico, with reporting two first records of A. theodorei and A. massartii. This diversity was evaluated with morphological and molecular analysis. The purpose of the study, the method commonly used, and the results presented are appropriately described in the MS. Thus, it can be published after minor revisions as follows.

Thank you for this indication of general support for publication after minor revisions

1.  Line 26 The dinoflagellate genus Amphidinium -> The genus Amphidinium

Agreed, this was redundant and has been deleted

2. Line 37 Herdmanii and Operculatum -> Italic

Here we disagree that these group names should be in italics – they are not species per se but rather informal clade designations. “As in all recent editions, scientific names under the jurisdiction of the Code, irrespective of rank, are consistently printed in italic type. The Code sets no binding standard in this respect, as typography is a matter of editorial style and tradition, not of nomenclature.” (Shenzhen Code, 2018: xxiv).

3. Line 182 50 mmol -> umol

Now corrected

4. In result section, It would be better if authors indicate the position of the nuclei in Photo images of each species.

This has been done but only for images where the position of the nucleus is evident; otherwise misleading to point to features not visible in the images.

5. In 3.1.3. Amphidinium massartii (AmLT112), In the discussion, it was explained that there is also the presence or absence of scale when distinguishing A. carterae and A. massartii by their morphological characteristics. Therefore, it would be nice to have data on the presence or absence of scale in the results of this paper.

The missing information on the presence/absence of scales is now added. A fragment “- details confirmed herein by SEM analysis” (lines 512-513) was deleted. We could not see scales using an SEM; TEM observations are necessary to distinguish scales provided they are present. Instead, a new fragment was added: Unfortunately, without observation under transmission electron microscopy (TEM), scales located outside the plasma membrane could not be distinguished. In scale-bearing Amphidinium species, scales are not visible by SEM because they are tiny, e.g. only 136.4 x 91.0 x 81.8 nm in A. cupulatisquama Tamura & Horiguchi (Tamura et al. 2009), 64.8-68.4 x 48.5-49.3 x 12.6-18.4 nm in two unidentified Amphidinium strains closely related to A. carterae (Sekida et al. 2019), and 276 ± 17 nm in A. stirisquamtum Z. Luo, Na Wang & H. Gu (Luo et 2021).

6. Line 414 Oper-414 culatum clade sensu lato -> Oper-414 culatum clade stricto

Here, we think the reviewer is referring to Line 415 (PDF file)? In that case, we believe it should remain as “sensu lato”, i.e. in the broadest, informal sense of the group affinity because these clades are only operationally defined.

Round 2

Reviewer 1 Report

I want to thank the authors for their efforts to make the manuscript better.

Now I have several small suggestions:

1. Lines 286, 468, 694 - (ITS1-5.8S rRNA rRNA-ITS2) > (ITS1-5.8S rRNA-ITS2).

2. Lines 325-345 - not bold please :)

3. Line 463 - Please provide references to the corresponding figures (S4 and S5)?

4. Line 466 - Please provide references to the corresponding figures (S2 and S3)?

5. Please, make sure that Fig 7 is still presented in the manuscript. There is a deleted one. Next two figs captions one by one.

6. Line 608, and below - please give the references to the tree(s) in every end of sentence.

7. Line 628 - levels [2,53].We > levels [2,53]. We

8. Line 626 - I'm not agree, that MP assumes a unique (parsimony) mutation model. It does not operate with models, it is a criterion, and under this criterion, the shortest possible tree that explains the data is considered best. Please, rephrase the sentence.

9. Please revise the Supplementary Materials section (lines 714-721). There is a repeat.

10. Please delete .1's at the Trees S4, S5.

Author Response

I want to thank the authors for their efforts to make the manuscript better

The authors appreciate the reviewer’s comments to improve the manuscript.

Now I have several small suggestions:

  1. Lines 286, 468, 694 - (ITS1-5.8S rRNA rRNA-ITS2) > (ITS1-5.8S rRNA-ITS2).

Done. This was an edition mistake and has been corrected. Thank you.

  1. Lines 325-345 - not bold please :)

This was changed in the edited journal version. Probably, the mistake was made in the journal’s platform. We have corrected this mistake.

  1. Line 463 - Please provide references to the corresponding figures (S4 and S5)?
  2. Line 466 - Please provide references to the corresponding figures (S2 and S3)?

References were provided for the full paragraph.

  1. Please, make sure that Fig 7 is still presented in the manuscript. There is a deleted one. Next two figs captions one by one.

The corrected manuscript previously submitted had this figure, but for unknown reasons, the manuscript was altered in the journal’s platform. In fact, in the journal’s version that you received, Fig 7 is placed behind Fig. 8. We are sorry for these errors. We are submitting again the correct version with all the figures.

  1. Line 608, and below - please give the references to the tree(s) in every end of sentence.

Done.

  1. Line 628 - levels [2,53].We > levels [2,53]. We

This word belonged to a different paragraph. Now it is corrected.

  1. Line 626 - I'm not agree, that MP assumes a unique (parsimony) mutation model. It does not operate with models, it is a criterion, and under this criterion, the shortest possible tree that explains the data is considered best. Please, rephrase the sentence.

Thanks for the observation; we agree with the reviewer, and the sentence was corrected.

  1. Please revise the Supplementary Materials section (lines 714-721). There is a repeat.

Done

  1. Please delete .1's at the Trees S4, S5.

Done.